# Learning Multimodal Rewards from Rankings

**Vivek Myers** [†]     **Erdem Bıyık** [‡]     **Nima Anari** [†]     **Dorsa Sadigh** [†,‡]

[†] Department of Computer Science, Stanford University
[‡] Department of Electrical Engineering, Stanford University
{vmyers,ebiyik}@stanford.edu, {anari,dorsa}@cs.stanford.edu

**Abstract:** Learning from human feedback has shown to be a useful approach in acquiring robot reward functions. However, expert feedback is often assumed to be drawn from an underlying *unimodal* reward function. This assumption does not always hold including in settings where multiple experts provide data or when a single expert provides data for different tasks—we thus go beyond learning a unimodal reward and focus on learning a *multimodal* reward function. We formulate the multimodal reward learning as a mixture learning problem and develop a novel *ranking*-based learning approach, where the experts are only required to rank a given set of trajectories. Furthermore, as access to interaction data is often expensive in robotics, we develop an *active* querying approach to accelerate the learning process. We conduct experiments and user studies using a multi-task variant of OpenAI's LunarLander and a real Fetch robot, where we collect data from multiple users with different preferences. The results suggest that our approach can efficiently learn multimodal reward functions, and improve data-efficiency over benchmark methods that we adapt to our learning problem.

**Keywords:** HRI, reward learning, multi-modality, rankings, active learning

## 1   Introduction

Learning a reward function from different sources of human data is a fundamental problem in robot learning. In recent years, there has been a large body of work that learns reward functions using various forms of human input, such as expert demonstrations [1], suboptimal demonstrations [2, 3, 4], pairwise comparisons [5, 6], physical corrections [7, 8], rankings [9], and trajectory assessments [10]. These works focus on learning a *unimodal* reward function that models human preferences on a target task. However, this unimodality assumption does not always hold: human preferences are usually more complex and need to be captured via a multimodal representation. Further, even if the preferences of a human are truly unimodal, we often use a mixture of data from multiple humans, which can be difficult to disentangle, leading to multimodality.

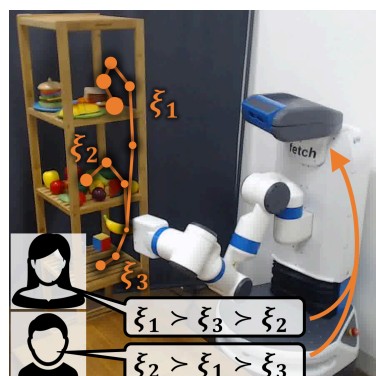

Figure 1: Fetch robot putting a banana on one of the three shelves. The two users have different preferences, and so they provide different rankings to the robot. The robot needs to be able to model multimodal reward functions for successfully achieving the task.

As an example, consider a robot placing a banana on one of the three shelves (see Fig. 1). The middle shelf is often used for fruits, but it has no room left and if the robot tries to put the banana there, it may cause other fruits to fall. The top shelf has some space but it has been used for cooked meals. The bottom shelf has a lot of free space, but is usually used only for toys. In such a scenario, people may have very different preferences about what the robot should do. If we try to learn a unimodal reward using data collected from multiple people, the robot is likely to fail in the task, because the data will include inconsistent preferences.

One solution is of course to label the different modes in the data. For example, one could separate the data based on the preferred shelf, and learn different reward functions for each shelf. However, this separation is not always straightforward. For example in a driving dataset, it is unclear what should be labeled as aggressive or timid driving. Clustering the data based on the human who provided

5th Conference on Robot Learning (CoRL 2021), London, UK.

the data is also not viable as it will introduce data-inefficiency issues, and perhaps more importantly, humans are not always unimodal: a usually timid driver can drive more aggressively when in a hurry.

These examples motivate us to develop methods that can learn *multimodal* reward functions using datasets that are not specifically labeled with the modes. To this end, previous work proposed learning from demonstrations to learn multimodal policies [11, 12] or reward functions with multiple possible intentions [13, 14]. However, learning from expert demonstrations is often extremely challenging in robotics as providing demonstrations on a robot with high degrees of freedom is nontrivial [5, 15], and humans have difficulty giving demonstrations that align with their preferences due to their cognitive biases [16, 17]. Thus, it is desirable to have methods that learn from other more reliable sources of data. For instance, humans can reliably compare two different trajectories, enabling a robot to learn from pairwise comparisons [5, 9, 18].

While learning from pairwise comparisons provides a rich source of data for learning reward functions, the theoretical results by Zhao et al. [19] imply that extending the existing comparison-based reward learning techniques to multimodal reward functions is not possible, i.e. failure cases can be constructed, where pairwise comparisons are not sufficient for identifying different modes of a multimodal function. Our insight is that it is possible to learn a multimodal reward function by going beyond pairwise comparisons and instead using *rankings*.

To achieve this, we formulate multimodal reward learning as a mixture learning problem. As data is a very expensive resource in robotics, we further develop an *active querying* method that aims to ask the human users the most informative queries. Our contributions are three-fold:

- We develop a method that uses rankings from humans to learn multimodal reward functions.
- We develop an active querying method to improve data-efficiency by asking the most informative ranking queries.
- We conduct extensive experiments and user studies with OpenAI's LunarLander and a real Fetch robot to test our learning and querying methods in comparison to baselines.

## 2  Related Work

**Reward Learning in Robotics.** Learning reward functions from human feedback is a fundamental problem in robot learning. Ng and Russell [20] and Abbeel and Ng [21] introduce the problem of learning from demonstrations in the space of robotics. Later works focus on improving *inverse reinforcement learning* algorithms by reducing the ambiguity in learned rewards [1, 22].

Due to the difficulty in providing expert demonstrations in robotics, recent works attempt to learn reward functions using other forms of human feedback, such as physical corrections [7, 8] and pairwise comparisons where a human user compares the quality of two robot trajectories [23, 24, 25, 26]. Later works extend these algorithms for better time and data-efficiency [27, 28, 29]. Though there have been works to extend this framework to reward functions modeled with Gaussian processes to capture nonlinearities [6, 30, 31], the underlying reward function has always been unimodal.

**Preference-based Learning.** Outside of robotics, preference-based learning, where data are in the form of comparisons, selections out of a set, or rankings, has attracted attention due to the ease of collecting data and its reliability. Prior works have studied this in the classification [32], bandits [33], and reinforcement learning settings [34].

In our problem, we have a continuous hypothesis space of reward functions. In such cases, it is common to model human comparisons or rankings with a computational model. Bradley-Terry is one such model for pairwise comparisons [35], which is easily extended to the queries where the human chooses the best of multiple items. Known as multinomial logits (MNL) [36], this model has been widely used for human preferences in many fields [37, 38, 39] including robotics [25, 27, 18].

To extend these models to rankings, Plackett-Luce [40, 41] and Mallows models [42, 43, 44, 45] are commonly employed. In this paper, we use the Plackett-Luce model as it is a natural extension of MNL, which is widely used in robotics with great success. We formalize this model in Section 3.

Even though there has been much research in this domain, all works we mentioned here focus on the unimodal case, and do not work with the multimodal preferences.

**Learning Mixture Models from Rankings.** One way to model multimodal reward functions is through mixture models, where the data is assumed to come from different individual models with some unknown probabilities. To this end, previous works consider mixtures of MNLs [46, 47],

Plackett-Luce models [19], and Mallows models [48]. Other works adopt different methods to model multimodality, such as by assuming latent state dynamics that transition between different modes [49, 50] or by learning the different modes from labeled datasets [51, 52]. To avoid these modeling assumptions, we focus on directly learning the mixture model.

While Zhao et al. [19] have theoretically studied the mixture of Plackett-Luce choice models, which also informs our algorithm in terms of the query sizes, they only focus on learning the rewards of a discrete set of items. To the best of our knowledge, our paper is the first work that deals with a continuous hypothesis space under a mixture of Plackett-Luce models. Furthermore, we propose an active querying strategy for this mixture model to improve data-efficiency for human-in-the-loop learning, which is crucial in data-hungry applications such as robotics.

## 3  Problem Formulation

**Setup.** We consider a fully-observable deterministic dynamical system. A trajectory $\xi$ in this system is a series of states and actions, i.e., $\xi = (s_0, a_0, \ldots, s_T, a_T)$. The set of feasible trajectories is $\Xi$.

We assume there is a set of $M$ individual reward functions that are possibly different, each of which encodes some preference between the trajectories in $\Xi$. For the rest of the formalism, we refer to each individual reward function as an *expert* for the clarity of the presentation.

Following the common linearity assumption in reward learning [1, 5, 28], we assume each preference can be modeled as a linear reward function over a known fixed feature space $\Phi$, so the reward associated with a trajectory $\xi$ with respect to the $m^{\text{th}}$ expert is $R_m(\xi) = \omega_m^\top \Phi(\xi)$, where $\omega_m$ is the unknown vector of weights. Across the expert population, there exists some unknown distribution over the reward parameters, corresponding to the ratio of the data provided by the experts. We represent this distribution with mixing coefficients $\alpha_m$ such that $\sum_{m=1}^M \alpha_m = 1$. We will then learn both the unknown reward functions $\{\omega_m\}_{m=1}^M$ and the mixing coefficients $\{\alpha_m\}_{m=1}^M$, using ranking queries made to the $M$ experts. This setup generalizes [5], which studied unimodal rewards.

**Ranking Model.** We define a *ranking query* to be a set of the form $Q = \{\xi_1, \ldots, \xi_K\}$ for a fixed query size $K$. The response to a ranking query is a ranking over the items contained therein, of the form $x = (\xi_{a_1}, \ldots, \xi_{a_K})$, where $a_1$ is the index of the expert's top choice, $a_2$ is the second top choice, and so on. While it is not known which expert provided the response to the query, we know the prior that a response comes from expert $m$ with some unknown probability $\alpha_m$, i.e., $\Pr(R = R_m) = \alpha_m$. Going back to our banana placing example, a ranking query of $K$ robot trajectories is generated by the algorithm, and a user—whose identity is unknown to the algorithm—responds to this query.

We then capture how human experts respond to these ranking queries by modeling a ranking distribution through an iterative process using Luce's choice axiom [53]. In this process, the experts repeatedly select their top choice $a_m$ with a probability distribution generated with the softmax rule to generate a ranking from the order items were selected:

$$\Pr\left(x_1 = \xi_{a_1} \mid R = R_m\right) = \frac{e^{R_m(\xi_{a_1})}}{\sum_{j=1}^K e^{R_m(\xi_{a_j})}} .$$

In the following iterations, the experts select their top choice among the remaining trajectories:

$$\Pr\left(x_i = \xi_{a_i} \mid x_1, \ldots, x_{i-1}, R = R_m\right) = \frac{e^{R_m(\xi_{a_i})}}{\sum_{j=i}^K e^{R_m(\xi_{a_j})}} . \tag{1}$$

This is known as the Plackett-Luce ranking model [40, 41]. Together with the prior over experts $\alpha_m$, the resulting distribution over rankings $x \sim X$ is a mixture of Plackett-Luce models with mixing coefficients $\alpha_m$ and weights proportional to $e^{R_m(\xi)}$.

Hence, the ranking distribution first selects the reward function $R_m$ with probability $\alpha_m$, and then selects trajectories from $Q$ sequentially with probability proportional to the exponent of their reward, i.e., $e^{R_m}$, among the remaining trajectories until none is left, generating a ranking of the trajectories.

So given knowledge of the true reward function weights $\omega_m$ and mixing coefficients $\alpha_m$, we have the following joint mass over observations $x$ from a query $Q$:

$$\Pr(x \mid Q) = \sum_{m=1}^M \alpha_m \prod_{i=1}^K \frac{e^{\omega_m^\top \Phi(\xi_{a_i})}}{\sum_{j=i}^K e^{\omega_m^\top \Phi(\xi_{a_j})}} . \tag{2}$$

**Objective.** Our goal is to design a series of adaptive queries $Q^{(t)}$ to optimally learn the reward weights $\omega_m$ and corresponding mixing coefficients $\alpha_m$ upon observing the query responses $x^{(t)}$. We constrain all queries to consist of a fixed number of elements $K$.

# 4 Active Learning of Multimodal Rewards from Rankings

In this section, we first start with presenting our learning framework. We then discuss how we can improve data-efficiency, and propose an active querying approach.

## 4.1 Learning from Rankings

To learn the reward weights $\omega_m$ and mixing coefficients $\alpha_m$, we adopt a Bayesian learning approach. For this, we maintain a posterior over the parameters $\omega_m$ and $\alpha_m$. Denoting the distribution over the parameters $\alpha_i$ and $\omega_i$ as $\Theta$, this posterior can be written as

$$\Pr(\Theta \mid Q^{(1)}, x^{(1)}, Q^{(2)}, x^{(2)}, \dots) \propto \Pr(\Theta)\Pr\left(Q^{(1)}, x^{(1)}, Q^{(2)}, x^{(2)}, \cdots \mid \Theta\right)$$

$$= \Pr(\Theta)\prod_t \Pr\left(x^{(t)}, Q^{(t)} \mid \Theta, Q^{(1)}, x^{(1)}, ..., Q^{(t-1)}, x^{(t-1)}\right) \propto \Pr(\Theta)\prod_t \Pr\left(x^{(t)} \mid \Theta, Q^{(t)}\right), \quad (3)$$

where we use the conditional independence of ranking queries $x^{(t)}$ given $\Theta$ and the conditional independence of the $Q^{(t)}$ on $\Theta$ given $Q^{(1)}, x^{(1)}, \dots, Q^{(t-1)}, x^{(t-1)}$ in the last equation. To be able to compute this posterior, we assume some prior distribution over the reward weights and the mixing coefficients, which is system-dependent and may come from domain knowledge, and use Eq. (2) to calculate the likelihood terms. For example, in our simulations and user studies, we adopted a Gaussian prior $\omega_i \sim \mathcal{N}(0, I)$ and a uniform prior $\alpha \sim \mathrm{Unif}(\Delta_{M-1})$ where $\Delta_{M-1}$ is the unit $M-1$ simplex. Learning this posterior distribution in Eq. (3), one can compute a maximum likelihood estimate (MLE) or expectation as the predicted reward weights and mixing coefficients.

Equation (3) implies the queries made to the experts, $Q^{(t)}$'s, affect how well the posterior will be learned. Assuming a limited budget of queries, which is often the case in many real-world applications, including robotics, one would ideally find an optimal adaptive sequence of queries such that the responses would give the highest amount of information about the reward weights and the mixing coefficients. However, this is NP-hard, even in the unimodal case with pairwise comparisons [54]. We therefore resort to greedy optimization techniques to develop our active learning approach.

## 4.2 Active Querying via Information Gain

A query $Q$ is desirable if observing its value $x$ yields high information about the underlying model parameters, $\alpha_m$ and $\omega_m$. Therefore, we propose using an information gain objective to adaptively select the most informative query at each querying step, generalizing the approach of [5].

Assume at a fixed timestep $t$ we have made past query observations $\mathcal{D} = \left\{Q^{(t')}, x^{(t')}\right\}_{t'=1}^{t-1}$. The desired query is then $Q^* = \arg\max_Q I(X; \Theta \mid Q, \mathcal{D})$ where $I(\cdot; \cdot)$ denotes mutual information. Equivalently, denoting the joint distribution over $x$ and $\theta = \{\alpha_m, \omega_m\}_{m=1}^M$ conditioned on $Q$ and $\mathcal{D}$ as $P(X, \Theta \mid Q, \mathcal{D})$, we see

$$Q^* = \arg\min_Q \ \mathbb{E}_{P(X, \Theta \mid Q, \mathcal{D})} \ \log \frac{\mathbb{E}_{\theta' \sim \Theta \mid \mathcal{D}} \Pr[X = x \mid Q, \theta']}{\Pr[X = x \mid Q, \theta]} \ . \quad (4)$$

The details of this derivation are presented in Appendix A.

## 4.3 Overall Algorithm

To efficiently solve the optimization in Eq. (4), we first note that we should use a Monte Carlo approximation since the expectations are taken over a continuous variable $\Theta$ and a discrete variable $X$ over an intractably large set of $K!$ alternatives. To perform this Monte Carlo integration, we require samples from the posterior $\Pr(X, \Theta \mid Q, \mathcal{D})$.

Our key insight is that we can obtain joint samples from both posteriors by first sampling from $\bar{\Theta} \sim \Pr(\Theta \mid \mathcal{D})$ and then sampling $x \sim \Pr\left(X \mid Q, \Theta = \bar{\Theta}\right)$ since $\Theta \perp Q \mid \mathcal{D}$ and $X \perp \mathcal{D} \mid Q, \Theta$. We perform the sampling $x \sim \Pr\left(X \mid Q, \Theta = \bar{\Theta}\right)$ efficiently using Eq. (2). In general, exact sampling from the posterior $\Pr(\Theta \mid \mathcal{D})$ is intractable. However, we note Eq. (3) can be directly evaluated (using Eq. (2)) and gives $\Pr(\Theta \mid \mathcal{D})$ up to a proportionality constant factor.

With this unnormalized posterior of Eq. (3), we use the Metropolis-Hastings algorithm as described in Appendix B to generate samples from the posterior $\Pr(\Theta \mid \mathcal{D})$.

We see our optimization problem simplifies to finding, for $N$ fixed samples $\bar{\theta}_i \sim \Pr(\Theta \mid \mathcal{D})$ and corresponding samples: $x_i \sim \Pr(X \mid Q, \bar{\theta}_i)$

$$\mathcal{L}(Q; x, \bar{\theta}) = \sum_{i=1}^{N}\left[\log\left(\sum_{j=1}^{N}\Pr[x_i \mid Q, \bar{\theta}_j]\right) - \log\Pr[x_i \mid Q, \bar{\theta}_i]\right], \quad Q^* = \arg\min_{Q} \mathcal{L}(Q; x, \bar{\theta}) . \quad (5)$$

We solve this optimization using simulated annealing [55] (see Appendix C). Algorithm 1 goes over the pseudocode of our approach, and we discuss the hyperparameters in our experiments in Appendix D.

---

**Algorithm 1** Active Querying via Information Gain

**Require:** Observations $\mathcal{D}$
1: $\{\bar{\theta}_i\}_{i=1}^{N} \sim \Pr(\Theta \mid \mathcal{D})$ w/ Eq. (3) via Metropolis-Hastings
2: **procedure** EVALQUERY($Q$)
3:     $\forall i, x_i \sim \Pr(x_i \mid Q, \bar{\theta}_i)$
4:         **return** $\mathcal{L}(Q; x, \bar{\theta})$                $\triangleright$ Eq. (5)
5: **end procedure**
6: $Q \leftarrow$ MINIMIZE(EVALQUERY)  $\triangleright$ Simulated annealing
7: select query $Q$

---

### 4.4 Analysis

We start the analysis by stating the bounds on the required number of trajectories in each ranking query to achieve *generic identifiability*. A Plackett-Luce model over $\Xi$ is generically identifiable if for any sets of parameters $\Theta_1$ and $\Theta_2$ inducing the same distribution over the responses to all queries of size $K$ on $\Xi$, the mixing coefficients of $\Theta_1$ and $\Theta_2$ are the same and the induced rewards $R_m(\xi)$ are identical across $\Xi$ up to a constant additive scaling factor.

**Theorem 1** (Zhao et al. [19]). *A mixture of $M$ Plackett-Luce models with query size $K$ and $|\Xi| = K$ is generically identifiable if $M \leq \lfloor\frac{K-2}{2}\rfloor!$.*

This statement follows directly from [19], which proves the above bound assuming that each query to the Plackett-Luce mixture is a full ranking over the set of items (i.e. $|\Xi| = K$). However, the assumption $|\Xi| = K$ is untenable in the active learning context, as it prevents any active query selection. To apply this result for our active learning algorithms, we relax the condition to $|\Xi| \geq K$.

**Corollary 1.1.** *A mixture of $M$ Plackett-Luce models with query size $K$ is* generically identifiable *if $M \leq \lfloor\frac{K-2}{2}\rfloor!$.*

We prove Corollary 1.1 in Appendix E.1. In our context, generic identifiability implies if the human response is modelled by a Plackett-Luce mixture, our Algorithm 1 will be able to recover its true parameters (up to a constant additive factor for the rewards) in the limit of infinite queries.

**Remark 1.** *Greedy selection of queries maximizing information gain in Eq. (4) is not necessarily within a constant factor of optimality.*

Appendix E.2 justifies Remark 1. In fact, greedy optimization of information gain for adaptive active learning can be significantly worse than a constant factor of optimality in pathological settings [56]. Despite its lack of theoretical guarantees, information gain is a commonly used effective approach in adaptive active learning [29, 57, 58]. Although other approaches like volume removal satisfy adaptive submodularity [25], they fail in settings with noisy observations by selecting high-noise low-information queries, and in practice achieve far worse performance than information gain.

## 5 Experiments

Having presented our learning and active querying algorithms, we now evaluate their performance in comparison with other alternatives. We start with describing the two tasks we experimented with:

**LunarLander.** We used 1000 trajectories in OpenAI Gym's LunarLander environment [59] shown in Fig. 2 (see Appendix G.1 for details on how they were generated).

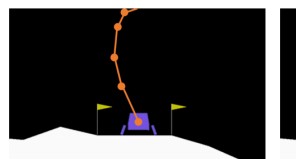 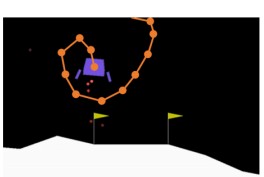

*(a) Landing softly on the landing pad*     *(b) Staying in the air as long as possible*

Figure 2: The LunarLander environment is visualized with the two tasks. Sample trajectories associated with these tasks are shown.

**Fetch Robot.** We generated 351 distinct trajectories of the Fetch robot [60] putting the banana on the shelves as shown in Fig. 1 (see Appendix G.2 for details).

### 5.1 Methods

We compare our active querying via information gain (IG) discussed in Algorithm 1 with two baselines: A simple benchmark for active learning is *random* query selection without replacement. We also benchmark against *volume removal* (VR), a common objective for active learning of robot reward functions [25]. See Appendix F for the details of these two baselines.

## 5.2 Metrics

We want to evaluate both the active querying and the learning performance. The former requires metrics that assess the quality of the algorithm's selected queries $\mathcal{D} = \left\{ (Q^{(t)}, x^{(t)}) \right\}_t$ in terms of the information they provide on the model parameters $\Theta$. We use two such metrics: mean squared error (MSE) and log-likelihood. Since both active and non-active methods are expected to reach the same performance with a large number of queries, we look at the area under the curve (AUC) of these two metrics over number of queries. To evaluate the learning performance, we quantify the success of a robot, which learned a multimodal reward, via the *learned policy rewards* on the actual task.

**MSE.** Suppose we know the human is truly modeled by $\Theta^*$ adhering to the assumed model class of Section 3. Given a set of observations $\mathcal{D}$, we can compute an MLE estimate $\widehat{\Theta}$ of the model parameters using Eq. (2). The MSE is then the squared error between $\widehat{\Theta}$ and $\Theta^*$ (see Appendix H.1).

While this metric cannot be evaluated with real humans, we can use this metric with synthetic human models (model with known parameters $\Theta^*$) in simulation. A lower MSE score means the selected queries $\mathcal{D}$ allow us to better learn a multimodal Plackett-Luce model close to the true model $\Theta^*$.

**Log-Likelihood.** The log-likelihood metric measures the log-likelihood of the response to a random query given the past observations $\mathcal{D}$. If the past observations $\mathcal{D}$ are informative, the true response to a random query $Q$ will in expectation be more likely, meaning the log-likelihood metric will be greater. See Appendix H.2 for details on how we compute this metric.

**Learned Policy Reward.** We take the MLE estimate of each reward weights vector and train a DQN policy using them [61].[1] We then run these learned policies on the actual environment with the corresponding true reward functions (see Appendix H.3) to obtain the learned policy rewards.

## 5.3 Results

**Multimodal Learning is Necessary.** We first compare unimodal and multimodal models to show the insufficiency of unimodal rewards when the data come from a mixture. To leave out any possible bias due to active querying, we make this comparison using random querying.

We let the true reward function have $M = 2$ modes and set a query size of $K = 6$ items for identifiability as Section 4.4 suggests, and for acquiring high information from each query. We simulate 100 pairs of experts whose reward weights $\omega_m$ and the mixing coefficients $\alpha_m$ are sampled from the prior $\Pr(\Theta)$. Having these simulated experts respond to 15 queries, we report the MSE in Fig. 3.

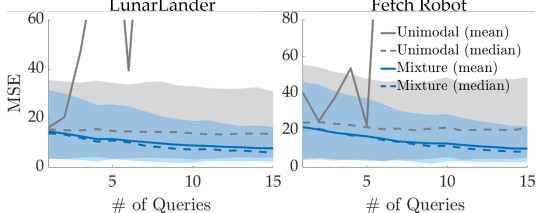

The unimodal reward model causes an unstably increasing MSE. This is mostly due to the outliers where the reward weights $\omega_1$ and $\omega_2$ are far away from each other and the unimodal reward fails to learn any of them. We therefore also plot the median values and quartiles in Fig. 3. While the bimodal reward model learned using our proposed approach decreases the MSE over time, the unimodal model has a roughly constant MSE, which suggests it is unable to learn when the data come from a mixture.

Figure 3: Unimodal and bimodal reward learning models are compared under MSE. Both mean and median values (over 100 runs) are shown. Shaded regions show the first and the third quartiles.

We present an additional unimodal learning baseline evaluated on the user study data in Appendix K.

**Active Querying with Information Gain is Data-Efficient.** We next compare our information-gain-based active querying approach with the other baselines. For this, we use the same experiment setup as above with $M = 2$ reward function modes and ranking queries of size $K = 6$, and simulate 75 pairs of human experts. We present the results in terms of MSE in Fig. 4. In LunarLander, the information gain objective significantly outperforms both random querying and volume removal in terms of the AUC MSE ($p < 0.005$, paired-sample $t$-test). Notably, volume removal performs even worse than the random querying method, which might be due to the known issues of volume removal optimization as briefly discussed in Appendix F.2. On the other hand, the difference is not statistically significant in the Fetch Robot experiment, which might be due to the small trajectory dataset,

---

[1] As we are using a real Fetch robot for our experiments and it would be infeasible and unsafe to train DQN on Fetch, so this metric is limited to our simulations, i.e., LunarLander in our experiments.

or because almost all trajectories in the dataset minimize or maximize some of the trajectory features, accelerating and simplifying learning under the linear reward assumption. See Appendix G.2 for details about the trajectory features and how we generated the trajectory dataset.

We further analyze the querying methods in this multimodal setting under the log-likelihood metric in Fig. 4. Information gain significantly outperforms random querying and volume removal in both experiments with respect to the AUC log-likelihood ($p < 0.005$). With respect to the final log-likelihood, information gain reduces the amount of required data in LunarLander by about 35% compared to random querying and about 60% to volume removal. Similarly in the Fetch Robot, the improvement is approximately 25% over both baselines.

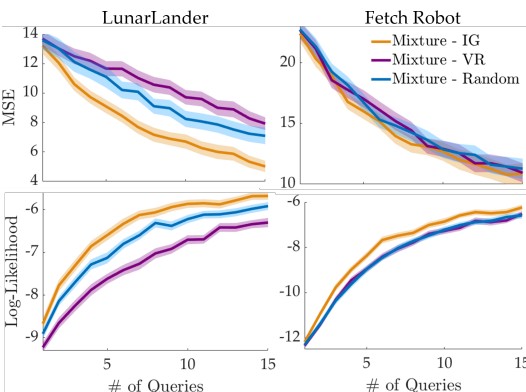

Appendix J presents two additional experiments: one which clearly shows the effectiveness of our approach for learning a mixture of more than two reward functions (specifically, $M = 5$), and one which studies the robustness against misspecified $M$.

Figure 4: Different querying methods are compared with the (top) MSE and (bottom) log-likelihood metrics (mean±se over 75 runs).

**Information Gain Leads to Better Learning.** Having seen the superior predictive performance of the reward learned via information gain optimization, we next assess its performance in the actual environment. As random querying outperforms volume removal in terms of log likelihood and MSE as in Fig. 4, we compare the information gain with random querying.

For this, we run the multimodal reward learning with 75 pairs of randomly generated reward weights ($M = 2$ and $K = 6$). For each of the 150 individual reward functions, we compute the learned policy rewards. Fig. 5 shows the results. While the standard errors in the plots seem high, this is mostly because optimal trajectories for different reward weights differ substantially in terms of rewards, which causes an irreducible variance. However, since the underlying true rewards are the same between the information gain and random querying methods, we ran the paired sample $t$-test between the results and observed statistical significance ($p < 0.05$). This means although the learned policy rewards between different runs differ substantially, the reward function learned via the information gain method leads to better task performance compared to random querying.

## 6 User Studies

We now empirically analyze the performance of our algorithm with two online user studies.[2] We again used the LunarLander and Fetch Robot environments. We provide a summary and a video of the user studies and their results at https://sites.google.com/view/multimodal-reward-learning/.

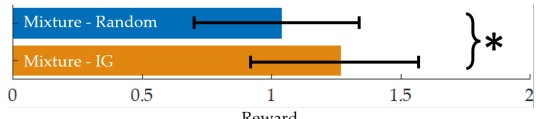

Figure 5: Information gain and random querying methods are compared with the learned policy rewards (mean±se over 75 runs which correspond to 150 randomly generated reward weights) in LunarLander.

**Experimental Setup.** For LunarLander, subjects were presented with either of the following instructions at every ranking query: "Land softly on the landing pad" or "Stay in the air as long as possible". We randomized these instructions such that users get one of them with 0.6 and the other with 0.4 probability. We kept the presented instructions hidden from the learning algorithms so that they need to learn a multimodal reward without knowing which mode each ranking belongs to.

For the Fetch robot environment, we recorded the 351 trajectories on the real robot as short video clips so that the experiment can be conducted online under the pandemic regulations. Human subjects participated in the experiment as groups of two to test learning from multiple users. Each participant was instructed that the robot needs to put the banana in one of the shelves and different shelves have different conditions (the same as in our running example, see Fig. 1, Appendix I.1).

After emphasizing there is no one correct choice and it only depends on their preferences, we asked each participant to indicate their preferences between the shelves on an online form. Afterwards,

---

[2]We have IRB approval from a research compliance office under the protocol number IRB-52441.

each group of two subjects responded to 30 ranking queries in total where each query consisted of 6 trajectories. We selected who responds to each query randomly, with probabilities 0.6 and 0.4.

Appendix I.2 presents details on the user interface used in our experiments.

**Independent Variables.** We varied the querying algorithm: active with information gain and random querying. We excluded the volume removal method to reduce the experiment completion time for the subjects, as it already performed worse than random querying in our simulations.

**Procedure.** We conducted the experiments as a within-subjects study. We recruited 24 participants (ages 19 – 56; 9 female, 15 male) for LunarLander and 26 participants (ages 19 – 56; 11 female, 15 male) for the Fetch robot. Each subject in the LunarLander, and each group of two subjects in the Fetch robot experiment responded to 40 ranking queries; 15 with each algorithm and 10 random queries for evaluation at the end. The order of the first 30 queries was randomized to prevent bias.

**Dependent Measures.** Learning the multimodal reward functions via the 15 rankings collected by each algorithm, we measured the log-likelihood of the final 10 rankings collected for evaluation.

**Hypotheses.** With LunarLander and Fetch robot, we test the following hypotheses respectively:
**H1.** *Querying the participants, who are trying to teach two different tasks, actively with information gain will lead to faster learning than random querying.*
**H2.** *While learning from two people with different preferences, active querying with information gain will lead to faster learning than random querying.*

**Results.** Figure 6 visualizes how log-likelihood of the evaluation queries changes over the course of learning by both algorithms. Active querying with information gain leads to significantly faster learning compared to random querying in LunarLander. Indeed, the difference in AUC log-likelihood is statistically significant ($p < 0.05$). Furthermore, the active querying method enabled reaching the final performance of random querying after only 9 or 10 queries, for around a 35% reduction in the amount of data needed, supporting **H1**.

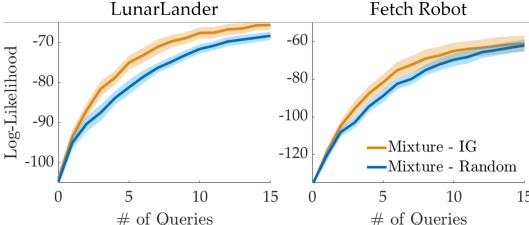

Figure 6: User study results (mean±se over 24 users for LunarLander and 13 groups for Fetch Robot).

As the robot experiments have a an easier task with a small number of variables between the trajectories, both querying methods converge to similar performances by the end of 15 queries. However, active querying accelerates learning in the early stages—the difference in AUC log-likelihood is again statistically significant ($p < 0.05$). Looking at the final performance with random querying, improvement in data efficiency is about 10%, supporting **H2**.

## 7 Conclusion

**Summary.** This work presents a novel approach for learning multimodal reward functions. We formulated the problem as a mixture learning problem solved using ranking queries that are answered by experts. We further developed an active querying method that maximizes information gain to improve the quality of ranking queries made to the experts. The results suggest our model learns multimodal reward functions, with data efficiency improved by our new active querying method.

**Limitations and Future Work.** Our model for learning multimodal rewards requires knowing the number of different modes (experts or tasks) $M$ in advance. This might be difficult in some settings. For example, when several experts belonging to different clusters, e.g., timid and aggressive drivers, provide data, it might be difficult to know the number of clusters in advance. However, a simple approach that fits the multimodal reward under various $M$ could reveal the true number of underlying modes. Another challenge is that learning a mixture reward model may contribute to the reward ambiguity problem in inverse reinforcement learning: each individual reward may have its own ambiguity. Future work should investigate the practical implications of this. In addition, theoretical results assert, to guarantee the reliable learning of a multimodal reward with $M$ modes, ranking queries should consist of $K$ queries such that $M \leq \lfloor \frac{K-2}{2} \rfloor!$. While this is manageable by multiple pairwise comparisons for each query or an iterative process where the expert selects the top item, it might consume too much time for large $M$. Thus, an interesting future direction is to investigate how to incorporate multiple forms of expert feedback, e.g., demonstrations in addition to rankings, to pretrain and reduce the required interaction time with humans.

**Acknowledgments**

The authors would like to acknowledge funding by NSF grants #1849952 and #1941722, FLI grant RFP2-000, and DARPA.

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
