# OpenReview forum: "Learning Multimodal Rewards from Rankings"
_robot-learning.org/CoRL/2021/Conference — CoRL2021 Oral_

### Official Review · Reviewer_94gh · 2021-07-04

**Originality:** Very Good
**Technical Quality:** Very Good
**Clarity Of Presentation:** Excellent
**Impact:** 4

**Recommendation:**

Strong Accept: I recommend accepting the paper and will argue for my recommendation even if other reviewers hold a different opinion.

**Summary:**

This paper proposes a method for multi-modal reward learning by formulating the problem as learning the parameters of a continuous mixture distribution based on ranked query responses by humans.


**Issues:**

See weaknesses section.

**Reviewer Expertise:**

Fair: Some knowledge of the area

**Strengths And Weaknesses:**

**Paper Strengths**
Writing: This paper is very well written, with good flow throughout the introduction and a clear explanation of the method. Furthermore, the experiments section is formatted well; the jumps from one subsection to the next made sense and the order in which results were presented makes the experiments feel intuitive.

Technical Contribution: The method is novel AFAIK, and it is a significant technical contribution with sensible design choices along the way.

Experiments: The experiments are comprehensive as they measure MSE, log likelihood, and perform real-world human studies on both a simulated environment and on a real robot. Furthermore, they justify the necessity of the design choices made for the method.

Results: The authors’ method shows decent to solid improvements over baselines across all of the experiments, thus demonstrating the superiority of the method.

**Paper Weaknesses**
Experiments: Although the simulated human feedback experiments demonstrate the need for multi-modal instead of unimodal reward models, results on user studies would still be elucidating.


**Summary Of Recommendation:**

The paper is a solid accept; experiments are robust and the method is a solid contribution.

---

> ### Author Response · Authors · 2021-08-28
> **Response to Reviewer 94gh**
>
> We thank the reviewer for their feedback and suggestions. We are in particular glad that they found it to be a well-written and significant technical contribution with comprehensive experiments. We address their primary concern regarding additional user study evidence that multimodal learning is necessary. We have made the changes in the paper with blue ink for the convenience of the reviewers.
>
> **Unimodal vs Multimodal Learning on Human Data:** Unfortunately comparing an active unimodal querying approach with a multimodal one would require additional human subject studies, which was not feasible during the rebuttal period. However, we did do additional analyses using the human subjects’ data collected via the random querying approach for this rebuttal. We summarize the methodology and discuss the results of these analyses below.
>
> Firstly, the meta review proposes an additional baseline of using the top ranked demonstration as the basis for learning a unimodal reward. We want to note that our problem setting requires learning a reward function from multiple rankings that possibly disagree (one for each expert query). Thus, there is no single top-ranked demonstration. Instead, there are many “top-ranked” demonstrations, one for each ranking generated by an expert. A key challenge that we address in our work is in fact reconciling these rankings that may disagree.
>
> A well-defined modification of this proposed baseline could be to select the unimodal reward that maximizes the sum of the reward across each of these top-ranked demonstrations. For this optimization to have a maximum, we must additionally constrain the baseline reward to have a fixed norm. By constraining the norm to be $1$, we have evaluated this baseline on the human experimental data with non-adaptive, random queries. We see it performs worse than using a bimodal MLE ($p=0.11$ for Fetch and $p=0.0001$ for Lunar Lander). We have reported these results in Appendix K.
>
> We note that for the synthetic data, the paper already presents an experiment establishing benefits to multimodal reward learning. This is shown in Figure 3.

---

> > ### Comment · Reviewer_94gh · 2021-09-03
> > **Response**
> >
> > Thanks for the reply!
> >
> > After reading the metareview, and the other reviews and your response/updated experiments, I still believe that this is a strong accept.

---

### Official Review · Reviewer_zn9H · 2021-07-22

**Originality:** Good
**Technical Quality:** Good
**Clarity Of Presentation:** Very Good
**Impact:** 3

**Recommendation:**

Weak Accept: I recommend accepting the paper, but will not argue for my recommendation if the majority of other reviewers have a different opinion.

**Summary:**

Paper proposes to learn multimodal reward functions where the datasets comprises trajectories generated with different intentions or goals. The proposed active learning method leverages rankings that are provided by the experts to better distinguish between the different modes. Theoretical justifications for the approach is given and the method worked well in the LunarLander and Fetch Robot experiments with faster convergence and lower MSE scores.

**Issues:**

Please see above regarding related work on MI-IRL and potential bias.

**Reviewer Expertise:**

Very good: Comprehensive knowledge of the area

**Strengths And Weaknesses:**

The paper is well-organized and clearly conveys the method and results. Using rankings to improve multiple-intention learning is new to my knowledge and is relevant to the robotics community, especially those working in LbD and/or IRL. The problem setting is realistic since datasets may not have goals/intentions explicitly labelled. The experiments and user study appear well-executed and show the approach is effective compared to volume removal and random query selection methods.

Although I'm positive about the paper, it does miss out on related work on multiple intention IRL [Babes-Vroman, 2011; Ramponi et al, 2020] where a similar mixture model was proposed. The key difference is that rankings are used in the learning process but these prior works should be discussed and compared against.

In Eqn (3), the term $Pr (x^{(t)} | \Theta, Q^{(t)})$ should be expanded and clarified. Since the queries depend on the previous rounds due to the inference, it's unclear whether (3) remains valid under the active inference setting? Would the estimated posterior be biased?

### References
- Monica Babes-Vroman, Vukosi Marivate, Kaushik Subramanian, and Michael L. Littman. Apprenticeship Learning About Multiple Intentions. In Proceedings of the 28th International Conference on Machine Learning, ICML ’11
- Giorgia Ramponi, Amarildo Likmeta, Alberto Maria Metelli, Andrea Tirinzoni, and Marcello Restelli. Truly Batch Model-Free Inverse Reinforcement Learning about Multiple Intentions. In International Conference on Artiﬁcial Intelligence and Statistics, 2020


**Summary Of Recommendation:**

Overall, the paper makes good technical contributions to the relevant problem of learning from a trajectory dataset with multiple intentions. I vote for acceptance.

---

> ### Author Response · Authors · 2021-08-28
> **Response to Reviewer zn9H**
>
> We thank the reviewer for their feedback and suggestions. We are glad that they found our paper to be well-organized, our problem to be realistic, and our experiment to show our approach to be effective. We address their primary concerns regarding additional references and mathematical clarity below. We have made the changes in the paper with blue ink for the convenience of the reviewers.
>
> **Additional References:** We thank the reviewer for their recommendation of additional related work on multimodal IRL from Babes-Vroman and Ramponi et al. We have changed our introduction section to cite and discuss this literature.
>
> **Posterior Bias in the Adaptive Setting:** The reviewer mentions concerns that selecting the queries $Q_i$ adaptively, and thus making them random variables, might bias the posterior in Equation 3. The intermediate steps in the derivation do indeed hold the $Q_i$ fixed in Equation 3. However, assuming that the queries are selected adaptively, the posterior in Equation 3 in fact still holds, since each query $Q_i$ is a deterministic function of the past queries $Q_1...Q_{i-1}$ and their responses $x_1...x_{i-1}$. Thus, intuitively, conditioning on the value of each subsequent query conveys no information in the adaptive setting that is not already conveyed by the past queries and their responses. We have modified the derivation of Equation 3 so that it clearly addresses this subtlety in the adaptive setting.

---

> > ### Comment · Reviewer_zn9H · 2021-09-01
> > **Thank you!**
> >
> > Thanks for your response to my queries and for amending the paper appropriately. With the change, I think the paper is good and acceptable for publication.

---

### Official Review · Reviewer_H6SU · 2021-07-24

**Originality:** Good
**Technical Quality:** Very Good
**Clarity Of Presentation:** Very Good
**Impact:** 3

**Recommendation:**

Weak Accept: I recommend accepting the paper, but will not argue for my recommendation if the majority of other reviewers have a different opinion.

**Summary:**

This paper addresses the problem of learning from multi-model ranking feedback from humans. Given that different people may have different preferences over a learning agent’s behavior, it is important to explicitly model this effect to capture the underlying data distribution. This paper also proposes an active learning algorithm that proposes ranking queries to maximize information gain.

**Issues:**

See weaknesses. More ablation experiments are needed.

**Reviewer Expertise:**

Good: General knowledge of the area

**Strengths And Weaknesses:**

To effectively and efficiently learn from human feedback is an important problem in robot learning. Learning from multiple humans adds additional challenges. This work proposes to explicitly learn multi-modal ranking models when we know there are different modes in the data. Experiments in simulation and with real human users demonstrate the effectiveness of the proposed active learning method.

One major weakness of the proposed method, as the authors also acknowledge, is that it requires knowing the number of different modes, which is unlikely to be true for many real applications. It is desirable to see results from simulation experiments to show how robust or sensitive the proposed algorithm is to a noisy estimation of the number of modes.


**Summary Of Recommendation:**

The problem setting is important, the proposed algorithm is novel and the experiments are convincing. However, the limitations of the algorithm is not well-tested. I recommend accepting the paper but will not argue against rejecting it.

---

> ### Author Response · Authors · 2021-08-28
> **Response to Reviewer H6SU**
>
> We thank the reviewer for their feedback and suggestions. We are glad that they found our problem interesting, our methods novel, and our experiments convincing. We address their primary concern regarding selection of the number of modes below. We have made the changes in the paper with blue ink for the convenience of the reviewers.
>
> **Robustness to M Parameter:** The reviewer mentions concerns about how to pick the $M$ parameter (number of modes), and wonders how in synthetic experiments performance varies when guessing different values for $M$. Conducting additional experiments on the complex environment described in Appendix J.2, using a 5 or 7 mode information gain agent is slightly better than a 3 mode agent which is in turn significantly better than a 1 mode information gain agent. We believe the new experiment and analysis in Appendix J.2 addresses these concerns, demonstrating relative robustness to the guessed value of $M$ as long as it is sufficiently large.

---

### Official Review · Reviewer_4NNL · 2021-07-26

**Originality:** Very Good
**Technical Quality:** Good
**Clarity Of Presentation:** Excellent
**Impact:** 3

**Recommendation:**

Strong Accept: I recommend accepting the paper and will argue for my recommendation even if other reviewers hold a different opinion.

**Summary:**

This paper addresses the problem of learning multimodal reward functions from interaction with a human teacher. The main contribution is the use of rankings, rather than just pairwise comparisons, to pose more informative queries to a teacher. Additionally, the paper presents an information-gain method for selecting optimal queries to pose to the teacher.

**Issues:**

* What is the benefit of learning a single model that is a mixture of multiple teachers' conflicting feedback? Rather than learning multiple models that reflect these preferences separately?
* What makes multiple experts' reward weights compatible enough that they can be represented using a single model? And at what point *should* they be represented using multiple models?
* How would this approach scale to M > 2 modes?

**Reviewer Expertise:**

Very good: Comprehensive knowledge of the area

**Strengths And Weaknesses:**

Strengths:
* Writing is clear and concise
* Ability to model conflicting feedback from teachers
* Use of human-provided data in the evaluation
* Information gain-based approach to query selection results in improved performance over baseline

Weaknesses:
* Multimodal approach is not well justified. More discussion is needed regarding the cases where a multimodal approach is preferable over learning multiple, unimodal models.
* Evaluation focuses mainly on testing the querying approach, rather than demonstrating the benefit of a multimodal approach.

**Summary Of Recommendation:**

This paper addresses the problem of learning multimodal reward functions from interaction with a human teacher. The main contribution is the use of rankings, rather than just pairwise comparisons, to pose more informative queries to a teacher. Additionally, the paper presents an information-gain method for selecting optimal queries to pose to the teacher.

The experiments described in this paper make use of data provide by real users in a user study, and the results indicate that the information gain method produces better results in a simulated lunar lander task and in a sorting task on a Fetch robot.

Overall, the goal of learning a multimodal model is an interesting technical challenge. However, the motivation for it is not yet clear. What is the benefit of learning a single model that is a mixture of multiple teachers' conflicting feedback? Rather than learning multiple models that reflect these preferences separately? The evaluation also focuses on the benefit of using an information gain approach to selecting queries, but does not provide a compelling justification for why a multimodal approach is ideal in the first place.

The implications of using a multimodal model should also be addressed. What makes multiple experts' reward weights compatible enough that they can be represented using a single model? And at what point *should* they be represented using multiple models? How would this approach scale to M > 2 modes?

Overall, this paper addresses a very interesting technical problem by modeling teachers' conflicting preferences in a single model. While the evaluation provides support for the proposed querying method, it does not yet fully justify why/when a multimodal approach should be used.

---

> ### Author Response · Authors · 2021-08-28
> **Response to Reviewer 4NNL**
>
> We thank the reviewer for their feedback and suggestions. We are excited to hear that they found our paper to be “clear and concise,” and thought that we addressed an “interesting technical challenge.” We will address the comments and concerns they brought up below. We have made the changes in the paper with blue ink for the convenience of the reviewers.
>
> **Justification for Multimodal Learning:** The reviewer mentions concerns that our experiments focus on the querying method more than the number of modes being learned. We note that the existing Figure 3 demonstrates that multimodal rewards can in certain situations perform better than unimodal rewards. The reviewer also wonders when learning additional modes is helpful and how our approach would scale to $M > 3$ modes. Conducting additional experiments on the complex environment as described in Appendix J.2, using a 5 or 7 mode information gain agent is statistically significantly better than a 3 mode agent which is in turn significantly better than a 1 mode information gain agent. This additional experiment shows our approach does in fact scale to $M > 3$ modes, and that for complex tasks, a large $M$ value can likely be used safely to robustly achieve good performance.
>
> **Learning Reward Functions Independently:** The reviewer questions why multimodal learning would be needed over learning multiple reward functions independently. We note that while we could learn multiple independent reward functions if we knew in advance which mode a human expert was sampling from, this assumption is often note true in practice. For instance, if expert feedback is submitted anonymously or depends on a non-public state (such as an expert's unstated, changing preferences), it would be difficult to separate rankings into different models to learn.

---

> > ### Comment · Reviewer_4NNL · 2021-09-03
> > **Response**
> >
> > Thank you for addressing my concerns, and for running the additional experiments on M>2 cases. I believe this added experiment strengthens the contributions of the paper, and I am changing my recommendation to "Accept".

---

### Meta-Review · Area_Chair_B4ix · 2021-08-16

**Recommendation:** Accept (Oral)
**Confidence:** 4

**Metareview:**

**Update after rebuttal**
I thank the authors for their thorough response, revising the paper based on feedback and clarifying some misunderstandings. My concerns have been addressed. I recommend accept

**initial meta-review**
**summary**
This work addresses the problem of learning reward functions from demonstrations that may have been generated from varying reward functions. It proposes an approach to leverage rankings over demonstrations to learn multi-modal reward functions. Furthermore, an active querying framework is introduced that aims to reduce the amount of demonstrations required.

**Strengths**

Learning reward functions from multiple different demonstrations is an important problem in the robotics community

A novel methodology of learning from rankings of demonstrations is presented, together with active query-ing approach that addresses scalability concerns

The approach is evaluated in simulated settings, as well as on demonstrations collected on real fetch robot.

**Weaknesses**

I agree with the reviewer asking for more motivation on why it’s important to learn multi-modal reward functions.  Additionally, I would expect additional baselines. For instance, a simple baseline could have been to just pick the top ranked demonstration and learn a uni-modal reward from that. This would have told us whether there is any benefit from learning multi-modal reward functions. Your current comparison (learning unimodal rewards from conflicting demonstrations vs learning multi-modal reward from conflicting demos) does not really answer the question of “are multi-modal rewards needed”. Without this additional baseline the impact of the proposed approach cannot really be measured.

The related work section should more clearly create connections between what has existed before, and how this work is extended. For instance currently the manuscript does not very clearly state which type of approach towards learning reward function weights is used, and what work your method is based on. Only the linear reward function assumption is stated, however various IRL styles use this assumption. After some searching, it seems like for instance this could be a good reference to clearly state as setting some foundations of the work in this manuscript: https://arxiv.org/pdf/2006.14091.pdf) . This lack of clearly stating what overall IRL approach is used makes it a) harder to understand the overall approach quickly b) harder to appreciate the progress made. Finally some other relevant related work should be included as pointed out by one of the reviewers.

The mixture of reward functions probably makes the issue of ambiguous reward functions worse? If yes it would be great to see some discussion on this topic, as it is typically a major concern of this type of IRL approach.

---

> ### Author Response · Authors · 2021-08-28
> **Response to AC**
>
> We thank the reviewers for their constructive comments and detailed feedback. We are happy that the AC thought that we are addressing an important problem and that our algorithms are novel. We will address comments that the reviewers have made relating to running additional baseline experiments and the necessity of multimodal learning. First, we will respond below to the concerns mentioned in the meta-review by the AC as well as the main concerns brought up by the reviewers. We have made the changes in the paper with blue ink for the convenience of the reviewers.
>
> **[AC] Additional Baseline:** The meta review proposes an additional baseline of using the top ranked demonstration as the basis for learning a unimodal reward. We want to note that our problem setting requires learning a reward function from multiple rankings that possibly disagree (one for each expert query). Thus, there is no single top-ranked demonstration. Instead, there are many “top-ranked” demonstrations, one for each ranking generated by an expert. A key challenge that we address in our work is in fact reconciling these rankings that may disagree.
>
> A well-defined modification of this proposed baseline could be to select the unimodal reward that maximizes the sum of the reward across each of these top-ranked demonstrations. For this optimization to have a maximum, we must additionally constrain the baseline reward to have a fixed norm. By constraining the norm to be $1$, we have evaluated this baseline on the human experimental data with non-adaptive, random queries. We see it performs worse than using a bimodal MLE ($p=0.11$ for Fetch and $p=0.0001$ for Lunar Lander). We have reported these results in Appendix K.
>
> We note that for the synthetic data, the paper already presents an experiment establishing benefits to multimodal reward learning. This is shown in Figure 3.
>
> **[AC] Additional References:** We thank the AC for pointing out the foundational role of Bıyık et al.’s work, which we had cited in the Related Works section. We have updated our problem formulation (Section 3) and active querying via information gain (Section 4.2) sections to cite and discuss this literature as well as the literature pointed out by the reviewer zn9H.
>
> **[AC] Ambiguous Reward Functions:** The AC mentions discussing the problem of ambiguous reward functions. We note that in Section 4.4 (Analysis), we present some discussion of this issue. In particular, we examine how existing Plackett-Luce identifiability literature can be applied to compute sufficient conditions for multimodal reward functions to be uniquely identifiable. However, while this identifiability condition makes sure each individual reward is recovered, it does not solve the reward ambiguity problem of IRL as the AC points out. That is, each individual reward may still suffer from the ambiguity problem: there might be many reward modes that explain the behavior of an expert. We have now added a brief discussion of this as a future work.
>
> **[4NNL] Justification for Multimodal Learning:** The reviewer mentions concerns that our experiments focus on the querying method more than the number of modes being learned. We note that the existing Figure 3 demonstrates that multimodal rewards can in certain situations perform better than unimodal rewards. The reviewer also wonders when learning additional modes is helpful and how our approach would scale to $M > 3$ modes. Conducting additional experiments on the complex environment as described in Appendix J.2, using a 5 or 7 mode information gain agent is statistically significantly better than a 3 mode agent which is in turn significantly better than a 1 mode information gain agent. This additional experiment shows our approach does in fact scale to $M > 3$ modes, and that for complex tasks, a large $M$ value can likely be used safely to robustly achieve good performance.
>
> **[4NNL] Learning Reward Functions Independently:** The reviewer questions why multimodal learning would be needed over learning multiple reward functions independently. We note that while we could learn multiple independent reward functions if we knew in advance which mode a human expert was sampling from, this assumption is often note true in practice. For instance, if expert feedback is submitted anonymously or depends on a non-public state (such as an expert's unstated, changing preferences), it would be difficult to separate rankings into different models to learn.

---

> > ### Author Response · Authors · 2021-08-28
> > **Response to AC (cont.)**
> >
> > **[H6SU] Robustness to $M$ Parameter:** The reviewer mentions concerns about how to pick the $M$ parameter (number of modes), and wonders how in synthetic experiments performance varies when guessing different values for $M$. Conducting additional experiments on the complex environment described in Appendix J.2, using a 5 or 7 mode information gain agent is slightly better than a 3 mode agent which is in turn significantly better than a 1 mode information gain agent. We believe the new experiment and analysis in Appendix J.2 addresses these concerns, demonstrating relative robustness to the guessed value of $M$ as long as it is sufficiently large.
> >
> > **[zn9H] Posterior Bias in the Adaptive Setting:** The reviewer mentions concerns that selecting the queries $Q_i$ adaptively, and thus making them random variables, might bias the posterior in Equation 3. The intermediate steps in the derivation do indeed hold the $Q_i$ fixed in Equation 3. However, assuming that the queries are selected adaptively, the posterior in Equation 3 in fact still holds, since each query $Q_i$ is a deterministic function of the past queries $Q_1...Q_{i-1}$ and their responses $x_1...x_{i-1}$. Thus, intuitively, conditioning on the value of each subsequent query conveys no information in the adaptive setting that is not already conveyed by the past queries and their responses. We have modified the derivation of Equation 3 so that it clearly addresses this subtlety in the adaptive setting.

---

### Decision · Program_Chairs · 2021-09-13

**Decision:**

Accept (Oral)

**Comment:**

**Update after rebuttal**
I thank the authors for their thorough response, revising the paper based on feedback and clarifying some misunderstandings. My concerns have been addressed. I recommend accept

**initial meta-review**
**summary**
This work addresses the problem of learning reward functions from demonstrations that may have been generated from varying reward functions. It proposes an approach to leverage rankings over demonstrations to learn multi-modal reward functions. Furthermore, an active querying framework is introduced that aims to reduce the amount of demonstrations required.

**Strengths**

Learning reward functions from multiple different demonstrations is an important problem in the robotics community

A novel methodology of learning from rankings of demonstrations is presented, together with active query-ing approach that addresses scalability concerns

The approach is evaluated in simulated settings, as well as on demonstrations collected on real fetch robot.

**Weaknesses**

I agree with the reviewer asking for more motivation on why it’s important to learn multi-modal reward functions.  Additionally, I would expect additional baselines. For instance, a simple baseline could have been to just pick the top ranked demonstration and learn a uni-modal reward from that. This would have told us whether there is any benefit from learning multi-modal reward functions. Your current comparison (learning unimodal rewards from conflicting demonstrations vs learning multi-modal reward from conflicting demos) does not really answer the question of “are multi-modal rewards needed”. Without this additional baseline the impact of the proposed approach cannot really be measured.

The related work section should more clearly create connections between what has existed before, and how this work is extended. For instance currently the manuscript does not very clearly state which type of approach towards learning reward function weights is used, and what work your method is based on. Only the linear reward function assumption is stated, however various IRL styles use this assumption. After some searching, it seems like for instance this could be a good reference to clearly state as setting some foundations of the work in this manuscript: https://arxiv.org/pdf/2006.14091.pdf) . This lack of clearly stating what overall IRL approach is used makes it a) harder to understand the overall approach quickly b) harder to appreciate the progress made. Finally some other relevant related work should be included as pointed out by one of the reviewers.

The mixture of reward functions probably makes the issue of ambiguous reward functions worse? If yes it would be great to see some discussion on this topic, as it is typically a major concern of this type of IRL approach.